# Team pro-social rule breaking and team innovation performance: An information processing theory perspective

Guosen Miao [1,2]*, Guoping Chen[1], Ying Yao[2]

1 Evergrande School of Management, Wuhan University of Science and Technology, Wuhan, Hubei, China,
2 Party School of Anhui Provincial Committee of C.P.C., Hefei, Anhui, China

* miaoguosen@wust.edu.cn

**Data Availability Statement:** All data files are available from Harvard Dataverse (https://dataverse.harvard.edu/dataset.xhtml?persistentId=doi:10.7910/DVN/UASKNB).

## Abstract

Exploring the antecedents that affect the team innovation performance can better promote the organization to research the potential factors to enhance the organizational innovation competitiveness. Drawing upon information processing theory, we develop a moderated mediation model to examine the relationship between team pro-social rule breaking and team innovation performance. A three-wave field study is constructed from two large manufacturing enterprises from 82 team leaders and their 382 subordinates in Shanghai, China. The results reveal that team pro-social rule breaking is positively related to team innovation performance through team reflexivity. In addition, the relationship between team pro-social rule breaking and team innovation performance via team reflexivity is positive only when team learning orientation is high. The implications, limitations, and future research directions of these findings are discussed.

## Introduction

The increasingly complex external environment and competitive markets of our current world call for organizations to adapt through innovation to gain a sustainable competitive advantage [1]; exploring the factors that impact the creativity of an organization's members is pivotal to both academia and organizations to find better ways to promote the improvement of organizational creativity performance. Employees, as drivers of creativity, inevitably challenge and break away from practices inherent in the workplace, potentially leading to the violation of rules and regulations deemed out of place. The behavior of employees who intend to violate a formal organizational regulation to promote the welfare of the organization or one of its stakeholders is defined as pro-social rule breaking, which has captivated scholars' attention in recent decades [2, 3]. Historically, researchers have focused on exploring the antecedents and consequences of individual pro-social rule breaking [4, 5], with many articles describing how individual characteristics and leadership style can stimulate employee pro-social rule breaking [6, 7] and how leader or employee pro-social rule breaking can lead to employee creative behavior or harm employee performance evaluation [8, 9]. Research has revealed the formation mechanism of individual pro-social rule breaking and its impact on organizations, but there is a lack of studies on team pro-social rule breaking. Exploration of the consequences of

**Funding:** The author(s) received no specific funding for this work

**Competing interests:** The authors have declared that no competing interests exist.

pro-social rule breaking at the team level could have more significant implications for organizations. Firstly, to respond effectively to the increasing dynamics and complexity of the environment, working as a team is the most common organizational strategy in all types of organizations today [10, 11], and the work of inter-team members influences each other to achieve common team goals. Existing studies on individual pro-social rule breaking lack the exploration of the impact of pro-social rule breaking on organizational outcomes in differentiated teams. Secondly, based on social information processing theory, team members will communicate with each other about ambiguous events to form a shared perception of the event [12–14]. These shared perceptions will further guide the individual's subsequent behavior. Pro-social rule breaking has an ambidextrous nature, which contains both pro-social and violating sides. Thus, team members are more likely to see it as an ambiguous event [2, 3, 15, 16]. The objective perception of organizational well-being that team members extract from such behaviors may encourage individuals to consider the organization's true goals and change their working patterns.

Based on social information processing theory, individuals will adjust their behavior within the organizational environment in which they study [12]. This adjustment's premise involves analyzing collected information [13]. Team members cannot automatically process information generated by ambiguous behavior of pro-social rule breaking. Instead, they use a systematic information-processing approach that triggers deeper thinking among team members about whether there are issues with the behavior or the rules. This, in turn, leads to a shared perception within the team. Consequently, team members can critically assess and reflect on whether the team's behavior is suitable for the current or intended environment, a process known as team reflexivity. Team reflexivity refers to the process by which team members openly reflect on team goals, strategies, and overall business processes to adapt to changes in the environment [17]. Previous studies have indicated that team reflexivity can encourage employees to develop an understanding of the organization's work goals by observing organizational factors such as the human resource system, leadership styles, and the diversity of team goals, ultimately leading to more innovative behaviors [17–21]. Although research on the influence of team reflexivity on creativity performance has unveiled the mechanism of team reflexivity on individual innovation investment, research on the external factors that trigger team reflexivity has primarily focused on the organizational and individual levels and has not explored the impact of team situational factors on team reflexivity [22]. As mentioned above, team members interact more frequently during working hours [10–12] and have more consistent work goals and clear work divisions [13, 14]. Consistent behavior patterns within a team, as an important source of organizational information, maybe more stimulating for individuals to reflect on work goals than other situational factors. Importantly, this reflective process involves the analysis of the current organizational situation and teamwork. In comparison with well-known psychological processes such as psychological contract breach [23] and leadership identification [24], this cognitive process can better emphasize the impact of social information processing on employees' cognition and behavior. Therefore, from the perspective of social information processing theory, this study posits that team factors, such as team pro-social rule breaking, may stimulate team members' reflexivity, further affecting team innovation performance. Conditionally, team reflexivity may serve as the mediating factor linking team pro-social rule breaking to team innovation performance.

With the logic above, the path of team reflexivity does not always proceed smoothly and necessitates certain environmental conditions. According to social information processing theory, the cognition of individuals within the team (e.g., team reflexivity) depends not only on past rule breaking but is also influenced by environmental cues [25]. As an important team behavior orientation, team learning orientation is an environmental condition with substantial information clues among various factors at the team level [26]. Team learning orientation

represents a high level of learning commitment and open-mindedness and encourages team members to identify, confront, and continually correct problems with existing rules or behaviors [27–29]. Thus, when organizations emphasize learning orientation, their members are more likely to openly question the irrational rules that govern organizational behavior within the team [30], which further enhances the team reflexivity process. Research on the influence of team learning orientation on creativity performance has indicated that organizational learning plays a pivotal role in enhancing organizational innovation capability and knowledge utilization [31, 32]. From a human resources perspective, the learning orientation work pattern embodies a set of methodologies for disseminating and applying knowledge within an organization, allowing for the transfer of acquired knowledge into the workplace, thus creating effective and efficient capabilities [33]. The research on organizational learning under various organizational scenarios has expanded scholars' insights into the interaction of team learning orientation and organizational characteristics on organizational output [34]. However, scholars' research on the interaction effects between team learning orientation and organizational characteristics in complex organizational contexts on organizational innovation output remains limited [30]. Existing studies cannot adequately explain whether an organizational atmosphere emphasizing learning can effectively stimulate individuals' deep understanding of organizational characteristics and subsequent organizational behaviors. In responding to the academic community's call for team learning-oriented research [30–32], we introduce team learning orientation as the boundary condition. The result will extend the research boundaries related to pro-social rule breaking and team learning orientation.

This study aims to make three primary contributions. Firstly, we endeavor to explore the positive effects of pro-social rule breaking on organizations at the team level. Scholars focus predominantly on the antecedents and consequences of pro-social rule breaking at the individual level, neglecting the definition and consequence analysis of pro-social rule breaking at the team level [2, 3, 6, 15, 35]. However, individual pro-social rule breaking is not isolated; it can potentially influence other team members' mental processes [36]. Therefore, it is pivotal to emphasize how team pro-social rule breaking affects the organization for future team management [37]. Concurrently, this study investigates the positive impact of team pro-social rule breaking on organizational innovation, expanding the research on the positive outcomes of pro-social rule breaking. Secondly, we consider team pro-social rule breaking as the antecedent of team reflexivity, broadening the research on organizational factors affecting team reflexivity. Utilizing team reflexivity as the intermediary linking team pro-social rule breaking to team innovation performance represents a valuable attempt to elucidate the path of team behavior influencing team performance within the social information processing theory framework. Thirdly, as a boundary condition, team learning orientation can effectively elucidate the impact of the interaction effect between team pro-social rule breaking and organizational factors on team reflexivity and team innovation performance in complex organizational contexts. This aligns with the academic demand for research into the trend of learning orientation [38]. The research findings will also establish a theoretical foundation for explaining the formation mechanism of diverse team innovation performance. The specific theoretical model is illustrated in Fig 1.

## Literature review and hypotheses

### Team pro-social rule breaking

Pro-social rule breaking reflects an individual's understanding of the working patterns of stakeholders and the organization's rules, but is there a consistent tendency for this pattern of behavior at the team level? Considering scholars' research on team organizational citizenship

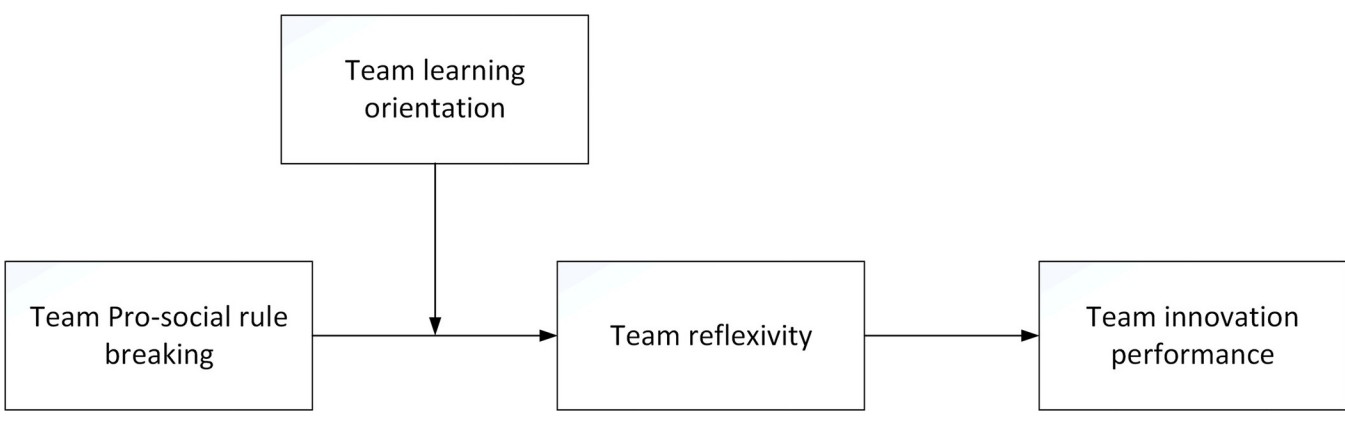

**Fig 1. Theoretical model.**

behavior and team organizational deviant behavior, we believe that aggregating individual pro-social rule breaking at the team level has certain theoretical support [39, 40]. Firstly, cognitive differences in pro-social rule breaking lead to inconsistent behaviors among different groups. Existing studies have demonstrated that varying social and environmental factors can result in different organizational behavioral trends [6, 11]. For instance, differences in leadership styles and human resource management systems can lead to varying levels of team reflexivity in different organizations [11, 17, 18]. Previous research has indicated that organizational atmosphere, job characteristics, and leadership styles, as social factors, influence individual pro-social rule breaking [6, 7, 15]. Although these factors may have different effects on individuals within an organization, they may form consistent patterns when aggregated at the team level. For example, individuals in inclusive organizations are more motivated to engage in pro-social rule breaking [35]. This conclusion is also applicable to specific regions. A study involving teachers in Uganda suggested that lower risk propensity leads to decreased motivation among teachers in that region to engage in pro-social rule breaking [16]. We posit that pro-social rule breaking may exhibit variations among teams under the influence of varying degrees of organizational factors rule breaking.

Furthermore, research on team organizational behavior also suggests that within a closely-knit organizational environment, team members may exhibit consistent behavioral patterns [41–45]. This consistency becomes significantly pronounced when team members share common tasks, job requirements, leadership, and organizational culture, which collectively foster a shared perception of the working environment [18, 40, 46–48]. As teams become more acquainted with the organization and each other, team members tend to develop a sense of organizational alignment [10, 29, 49, 50], which leads to similar behaviors [43, 45]. We contend that when team members encounter organizational factors that promote the formation of pro-social rule breaking, heightened internal communication will result in the alignment of internal members' actions and ultimately lead to the unity of internal behaviors.

## The manipulation of team pro-social rule breaking

Team-level concepts can be categorized into global, shared, and configural [51]. The global and configural types emerge through aggregating individual-level concepts at a higher level [45]. Additionally, the shared type can form through the composition of individual-level concepts where the two levels are similar [51]. Therefore, team pro-social rule breaking can be classified as a shared-type concept and is isomorphic to the concept of individual-level pro-social rule breaking.

Typically, operational methods for team-level concepts include the mean, variance, and maximum/minimum value methods. The variance method is suitable for exploring the diversity of team variables, such as the age diversity of team members. The maximum/minimum method is appropriate when team outcomes rely entirely on a specific individual within the team [52]. Moreover, the concept of team pro-social rule breaking primarily focuses on the prevalence of pro-social rule breaking within the team rather than on the diversity of individual team members' pro-social rule breaking behaviors. Both pro-social rule breaking behavior and voice behavior fall under the category of positive organizational behaviors and share similar risk-taking characteristics. In line with related research on team voice behavior [46], we utilize the mean method to operationalize the concept of team pro-social rule breaking. In other words, we use the mean value of all team members' pro-social rule breaking behavior to measure team pro-social rule breaking.

## Team pro-social rule breaking and team innovation performance

Researches indicate that pro-social rule breaking, which reflects individuals' autonomy in work and their investment in others' resources and aims to maximize stakeholders' needs, is often related to positive organizational performance [3, 4, 8]. Based on existing research, team pro-social rule breaking may also be a direct factor influencing team innovative performance.

Firstly, pro-social rule breaking indicates that employees, to a certain extent, overlook organizational norms in pursuit of work efficiency [2]. This behavioral characteristic reflects employees' ability to freely choose more appropriate methods to complete tasks when faced with conflicts between behaviors and norms. Research by Chang (2023) and other scholars has indicated that a high level of work autonomy is often associated with team innovation [41]. At the team level, the overt characteristics of pro-social rule-breaking behavior can facilitate more efficient organization of work interactions among colleagues, which will be beneficial for the improvement of team innovative performance [18]. Secondly, pro-social rule breaking also embodies assistance towards colleagues, manifested as a common practice of helping others within the team [2]. Existing research has shown that helping behavior within teams promotes the enhancement of team innovative performance [19]. Additionally, similar to efficiency-oriented pro-social rule-breaking behavior, pro-social rule-breaking behavior towards customers can convey the true purpose of work to individuals and strengthen individuals' awareness of work autonomy [2]. From an observer's perspective, individuals' perceptions of colleagues' pro-social rule-breaking behavior towards customers can provide specific service inspirations, and these cues can provide external perceptual conditions for individual innovation from aspects such as interaction methods and work processes [9], contributing to the improvement of collective innovative performance at the team level. In conclusion, a hypothesis regarding team pro-social rule-breaking behavior and team innovative performance is proposed: H1: Team pro-social rule-breaking behavior positively influences team innovative performance.

H1: Team pro-social rule-breaking behavior positively influences team innovative performance.

## Team pro-social rule breaking and team reflexivity: The moderation of the team learning orientation

As a group phenomenon, team pro-social rule breaking refers to the extent to which the behavior of most team members deviates from organizational rules. According to the social information processing theory, as an individual capable of self-regulation [53], an employee's subsequent behavior depends on past behavior and is influenced by environmental cues. The

complex and diverse organizational environment increases employees' confusion about the true intentions of the organization [28]. To resolve these ambiguities, individuals will analyze unique factors in the organizational context to deepen their understanding of the organization [29, 30]. Therefore, we argue that in the context of high team learning orientation, organizations encourage employees to deepen their understanding of organizational work goals through learning and encourage employees to optimize their work patterns through learning [43, 54]. In this context, employees are more likely to view team pro-social rule breaking as clues to the organization's intentions. Team learning orientation encompasses a set of values that influence the creation and utilization of knowledge within a team, shaping the team's learning atmosphere [31, 32, 38]. It can influence an organization in three key aspects: learning commitment, open thinking, and shared vision [27, 55]. Studies have demonstrated that organizations with a high-level team learning orientation consider differential team behavior a valuable learning source [29, 30, 56, 57]. Therefore, common behaviors that violate rules to improve team performance or assist others are likely to stimulate further reflection among team members in a team with a high-level team learning orientation. In other words, whether this phenomenon, reflecting the disparity between behaviors and rules, can trigger team reflexivity depends on the extent of team learning orientation.

Pro-social rule breaking is more likely to promote team reflexivity as a team with a high-level learning orientation. First, high learning commitment implies that it is important for team members to discern the implicit reasons and effects of explicit behaviors [47]. They may delve into the nature of pro-social rule breaking and potentially identify the conflict between the behavior and existing rules and regulations, uncovering the pro-social motivations behind the behavior. Since employees' pro-social rule breaking behavior does not conform to the organization's regulations, team members cannot automatically process such behavioral information; instead, they engage in systematic processing [58], which entails prompting the team to consider the conflict between behavior and rules deeply. Second, a team with high learning commitment and open thinking encourages members to identify and continuously optimize existing rules or address behavioral issues [27], breaking free from "fixed thinking modes." Furthermore, pro-social rule breaking falls under constructive deviant behavior [59]. Teams characterized by a strong learning orientation view employees who engage in constructive deviant behavior as positive change agents within the organization [60]. They actively share resources with others and may establish trusting relationships to gain understanding and support from others [61]. Therefore, a team with a higher level of learning orientation indicates that team members are more able to trust each other, stimulate a sense of cooperation among themselves, and engage in candid analysis and exchange of diverse views and opinions regarding the relationship between group behavior and organizational rules. Finally, a shared vision clarifies the team's learning direction and provides consistent goals for team members [57]. Consequently, team members develop a stronger sense of a "community of common destiny." They are more willing to actively discuss and reflect upon the conflicts between rule breaking behaviors and organizational rules with other members [19].

Conversely, on the one hand, in a team with a low-level learning orientation, team members may lack the motivation to delve into the implicit reasons and effects of explicit behavior [62]. Thus, they may struggle to uncover the pro-social motivations behind rule breaking behavior and consider it a common violation. On the other hand, team members might find it challenging to summon the courage to question the validity of existing rules. Moreover, uniting as a team can be difficult without a shared vision [63].

In summary, low-level team learning orientation makes it difficult for the entire team to gather clues from the environment and interactions with others, which are necessary to support open discussions and reflections on team behavior [64]. Therefore, motivation and

essential components for team reflexivity are lacking, resulting in an incomplete examination of the issues existing within the team [65]. In conclusion, team members with low-level learning orientation tend to disregard the cues that could otherwise stimulate open discussions about team behavior. This leads to a need for more motivation and key elements necessary for team reflexivity. Consequently, the following hypotheses are proposed.

H2: The positive relationship between team pro-social rule breaking and team reflexivity is moderated by team learning orientation. The relationships are stronger for team learning orientation is high (vs low).

## Team reflexivity and team innovation performance

Social information processing theory highlights that individuals' understanding and interpretation of situations can guide their related behaviors [66]. Social information cues can influence individuals' identification of organizational goals [12], thus impacting their subsequent behaviors [13]. It has been noted that a team, functioning as a complete information processing system, can establish team goals, processes, and outcomes through activities like reflection [17]. Specifically, a team with a high level of reflexivity can thoroughly contemplate team goals, strategies, and processes [19]. Such teams are inclined to swiftly identify and reflect upon issues related to processes, goals, and strategies, subsequently making effective predictions about external environmental changes and decisions. This, in turn, enhances the innovation performance of the entire team [20, 21, 62, 67].

Firstly, when a team encounters a high degree of uncertainty, team reflexivity can facilitate better integrating of external information cues into the team's cognitive framework and more effectively utilize existing resources to address difficulties and challenges. This helps the team innovate its plans, actions, and other processes, aligning team activities more closely with the evolving situation [20].

Secondly, a team with a high level of reflection encourages employees to express their ideas, giving due consideration to different opinions in comprehensive discussions [67]. This is pivotal for improving the innovative thinking of team members and enhancing the quality of the team's decision-making [21]. Open communication reduces differences in team members' task understanding, thus mitigating the negative impact on teamwork performance [68].

Thirdly, a team with high reflexivity fulfills the organizational system's need for constructive thinking, enabling team members to promptly identify key factors [48, 69]. Concurrently, it is essential to continuously integrate the skills of team members and organize them more effectively to coordinate their work [70].

Finally, through team reflexivity, employees can openly express their views on the team's internal and external aspects and actively discuss the team's shortcomings [22]. Team members will share their valuable views, contributing to disseminating new knowledge among members [11]. The team can continuously adjust and absorb new knowledge and information in an ever-changing external environment through reflexivity, generating more reasoned ideas and fostering innovation [18].

Taking together, the study puts forward the hypothesis as follows:

H3: Team reflexivity is positively related to team innovation performance.

Social information processing theory explains the impact of organizational factors on employee behavior. Given the context of this study, the interaction between team pro-social rule breaking and team learning orientation, as significant organizational cues, stimulates employees to recognize their expectations and job requirements within the organization. This

process is internalized through team reflexivity, becoming a driving force for employees' innovation in their work. Existing research indicates that social information processing theory explains how organizational behavior influences employee behavior [66]. Individual expectations of their work patterns are formed by analyzing continuous or isolated organizational behaviors, guiding them to improve their work output accordingly [12]. This behavioral pathway is influenced by more organizational behavior in complex organizational environments since employees face more significant difficulties understanding organizational intentions. Consequently, they rely more on information provided by the organization to analyze its needs [33]. Team learning orientation enhances the learning motivation of individuals within the organization [28]. Employees are more willing to improve their work performance by learning from the behavior of others [29]. A high level of team learning orientation promotes individual understanding of team pro-social rule breaking through internal reflection and discussion [32]. Through concentrated internal discussions, individuals within the team can form a consensus on innovative work patterns, further guiding the improvement of team innovation performance [31]. Based on the above analysis, team learning orientation moderates the indirect relationship between team pro-social rule breaking and team innovation performance through team reflection. The higher the team learning orientation, the stronger the indirect relationship. Therefore, this study proposes the following hypotheses:

H4: Team learning orientation moderates the indirect relationship between team pro-social rule breaking and team innovation performance through team reflexivity, such that the negative indirect relationship between team pro-social rule breaking and team innovation performance is stronger when employees with higher team learning orientation than when it is lower.

## Materials and methods

### Sample and procedure

Two large manufacturing enterprises from the Yangtze River Delta region were recruited to participate in a field survey. The first company is a large motor company headquartered in Shanghai, China, and the other company is a well-known enterprise in China's heavy equipment manufacturing industry headquartered in Shanghai. To reduce common method variance and illusory correlations, we collected data in three waves between June 1st and September 1st in 2019. According to the research topic, we selected full-time staff in sales, production, administration, and other departments as survey participants.

Due to the uncertainty of employee access to computers and to ensure an adequate volume of data, we used paper-based surveys during the first wave of data collection. We conducted a questionnaire survey for all employees, which included the pro-social rule breaking scale to be filled in by employees and the team learning-oriented scale to be filled in by supervisors. Employees and supervisors also reported their personal information, such as age, gender, and education level. We also clarified the research's purpose, emphasizing that data would solely be used for scientific research. More importantly, we encouraged participants to provide valid email addresses or WeChat IDs (a popular Chinese mobile messaging app) for active participation in the subsequent online phases of the study. After the survey, we provided each participant with approximately 40 RMB (It's worth about 5 dollars) in cash and promised an additional 80 RMB (It's worth about 10 dollars) upon completing the final wave of surveys.

With the assistance of the Wenjuanxing website (https://www.wjx.cn), the Chinese version of Qualtrics, each questionnaire was assigned a unique ID automatically generated within Wenjuanxing. Utilizing the collected information from the paper-based questionnaires, we

established a one-to-one correspondence between the questionnaire number, work number, email address, and Wenjuanxing ID, recording them for reference.

Approximately one-month later (time 2), we employed the online survey website for the team reflexivity questionnaire, which supervisors completed. Another month later, in time 3, we sent out the third survey questionnaire online, requesting supervisors to evaluate team innovation performance.

Initially, we distributed the first questionnaire link to 500 subordinates and 102 supervisors. During time 2, 417 participants who provided valid email addresses or WeChat IDs completed the second stage survey. One-month after that (time 3), the 417 participants were asked to complete the third stage questionnaire. Ultimately, 382 participants from 82 groups completed all three stages of the survey, resulting in a total response rate of 76.4%. Each group consisted of three to seven employees. Among the 382 employees, 50% were women, 86.91% were aged 39 or younger, 42.93% possessed a bachelor's degree or higher, and the average organizational tenure was 2.5 years.

## Measure

All measures were in Chinese and followed the standard translation–back translation process [71]. The participants completed the measures using a five-point Likert scale unless otherwise noted (1 = strongly disagree to 5 = strongly agree).

Pro-social rule breaking. Pro-social rule breaking was measured at time 1 using the 13-item scale developed by Dahling et al. [3]. Sample items include "I break company rules and regulations to save company time and money" and "I would break company rules and regulations if my co-workers need help at work." Cronbach's alpha for the scale was 0.926.

Team learning orientation. We asked the participants to rate the level of team learning orientation with their supervisors at time 2. Team learning orientation was measured using the 11-item scale developed by Sinkula et al. [27]. Sample items include "Learning is seen as the key to continuous team improvement" and "Employees and managers in the team have a common vision of the organization consensus." Cronbach's alpha for the scale was 0.984.

Team reflexivity. We used the 11-item team reflexivity scale developed by Schippers et al. [17]. The participants rated the scale at time 2 to measure the extent of their team reflexivity. Sample items are "Teams often summarize their work experience." Cronbach's alpha for the scale was 0.939.

Team innovation performance. We used the six-item team innovation performance scale developed by Prajogo et al. [72]. The scale was rated by the supervisors at time 3. A sample item is "Our team develops a wider variety of innovative products." Cronbach's alpha for the scale was 0.919.

Control variables. Previous empirical studies have confirmed that demographic variables such as gender, age, and education level can affect employees' pro-social rule breaking. This study controlled for employee age, education level, and income level [3]. In relation to team innovation performance and team reflexivity [62, 73], we controlled the percentage of female and team size for their potential influence on team innovation performance and team reflexivity. According to Wang's suggestion, our research also controlled team tenure because it takes time to establish the relationship between subordinates and supervisors [35].

## Method

The statistical tools SPSS21.0 and Mplus8.3 were used to analyze the data. Descriptive statistical analysis and hierarchical regression analysis were used to test hypothesis 1 and hypothesis 2. Furthermore, for testing hypothesis 3, considering the small sample size of the team, the

bootstrapping method can reduce the deviation caused by a small sample size. Hence, according to the suggestion from Preacher et al., we used bootstrapping confidence intervals to analyze the moderated mediation model [74]; the bootstrapping sample size was 20,000.

**Data aggregation.** The measurement of team pro-social rule breaking was aggregated through individual data. In order to prove the aggregation effect of the data of team pro-social rule breaking, referring to previous studies [75, 76], we first calculated the $r_{wg}$ of team pro-social rule breaking. The results showed that the $r_{wg}$ of team pro-social rule breaking meets the study requirements (average = 0.905; median = 0.952) [75]. Similarly, the intra-class correlation coefficient (ICC (1)) is 0.370, showing the proportion of variance interpreted by team members [77]. To assess the group effect, we also tested the ICC (2) of team pro-social rule breaking, which was 0.669, indicating that the proportion of variance explained at the team level met the study requirements [78]. These results support the summarization of individual scores for all target variables at the team level.

## Results

### Descriptive statistical analysis

The means, standard deviations, and correlations for each variable studied are provided in Table 1. The results show that team reflexivity has significant and positive relations with team innovation performance (r = 0.315, p <0.01). Thus, the correlation results align with theoretical expectations and provide a basis for further analysis. The results also show that team pro-social rule breaking has no significant relations with team innovation performance (r = 0.092, p >0.05), Hypothesis 1 was not supported.

### Reliability and validity

We used Cronbach's alpha coefficients to assess the reliability of the scales. The Cronbach's alpha for all constructs is between 0.919 and 0.984, which exceeds the recommended minimum standard value [79]. Moreover, according to Table 2, the composite reliability (CR) of variables is between 0.921 and 0.984, which is in line with the recommendations of scholars [79]. Therefore, the reliability of the measurements is acceptable.

**Table 1. Descriptive statistics and correlations among all variables.**

|  | Mean | SD | 1 | 2 | 3 | 4 | 5 | 6 | 7 | 8 | 9 | 10 |
|---|---|---|---|---|---|---|---|---|---|---|---|---|
| 1.edu | 3.090 | 0.888 |  |  |  |  |  |  |  |  |  |  |
| 2.incoming | 7.040 | 3.844 | 0.669** |  |  |  |  |  |  |  |  |  |
| 3.age | 33.100 | 3.847 | -0.126 | 0.120 |  |  |  |  |  |  |  |  |
| 4.tenure | 30.050 | 21.302 | 0.006 | 0.097 | 0.266* |  |  |  |  |  |  |  |
| 5.team size | 4.660 | 0.741 | -0.009 | 0.084 | 0.051 | -0.168 |  |  |  |  |  |  |
| 6. Percentage of females | 0.555 | 0.273 | -0.181 | -0.175 | -0.202 | -0.284** | -0.007 |  |  |  |  |  |
| 7.TPSRB | 2.186 | 0.530 | 0.058 | -0.061 | -0.311** | -0.244* | 0.120 | 0.107 | (0.926) |  |  |  |
| 8.TLO | 3.582 | 0.910 | -0.179 | -0.043 | -0.046 | -0.031 | 0.129 | 0.095 | 0.004 | (0.984) |  |  |
| 9.TR | 3.787 | 0.558 | -0.112 | -0.012 | -0.123 | -0.073 | 0.191 | 0.104 | 0.201 | 0.615** | (0.939) |  |
| 10.TIP | 3.254 | 0.848 | -0.104 | -0.101 | -0.151 | -0.008 | 0.091 | 0.107 | 0.092 | 0.317** | 0.315** | (0.919) |

Note: *p<0.05

**p<0.01, the alpha confidence coefficient of each variable is shown in parentheses.

Abbreviations: TPSRB, team pro-social rule breaking; TLO, team learning orientation; TR, team reflexivity; TIP, team innovation performance.

**Table 2. Convergent validity, CR and AVE.**

| Variable | items | loadings | CR | AVE |
|---|---|---|---|---|
| team pro-social rule breaking | item1 | 0.587 | 0.927 | 0.496 |
| | item2 | 0.662 | | |
| | item3 | 0.737 | | |
| | item4 | 0.652 | | |
| | item5 | 0.598 | | |
| | item6 | 0.645 | | |
| | item7 | 0.713 | | |
| | item8 | 0.784 | | |
| | item9 | 0.761 | | |
| | item10 | 0.787 | | |
| | item11 | 0.78 | | |
| | item12 | 0.704 | | |
| | item13 | 0.702 | | |
| team reflexivity | item1 | 0.81 | 0.942 | 0.598 |
| | item2 | 0.852 | | |
| | item3 | 0.793 | | |
| | item4 | 0.81 | | |
| | item5 | 0.691 | | |
| | item6 | 0.721 | | |
| | item7 | 0.824 | | |
| | item8 | 0.856 | | |
| | item9 | 0.648 | | |
| | item10 | 0.669 | | |
| | item11 | 0.793 | | |
| team learn-orientation | item1 | 0.882 | 0.984 | 0.85 |
| | item2 | 0.884 | | |
| | item3 | 0.968 | | |
| | item4 | 0.938 | | |
| | item5 | 0.973 | | |
| | item6 | 0.952 | | |
| | item7 | 0.943 | | |
| | item8 | 0.895 | | |
| | item9 | 0.969 | | |
| | item10 | 0.854 | | |
| | item11 | 0.875 | | |
| team innovation performance | item1 | 0.722 | 0.921 | 0.66 |
| | item2 | 0.788 | | |
| | item3 | 0.908 | | |
| | item4 | 0.795 | | |
| | item5 | 0.818 | | |
| | item6 | 0.832 | | |

We employed the Fornell–Larcker criterion to assess construct discriminant validity. The square root of AVE (average variance extracted) ranged from 0.702 to 0.922, while correlation coefficients between the variables ranged from 0.004 to 0.615. In all cases, the square root of AVE exceeded the correlation coefficients between variables, providing confirmation of discriminant validity [80].

**Table 3. Results of confirmatory factor analysis.**

| Model | $\chi^2$ | df | $\chi^2/df$ | TLI | CFI | RMSEA | SRMR | $\Delta\chi^2$ |
|---|---|---|---|---|---|---|---|---|
| four-factor model | 173.277 | 48 | 3.610 | 0.962 | 0.972 | 0.083 | 0.036 | |
| three-factor model 1 | 998.734 | 51 | 19.583 | 0.730 | 0.791 | 0.221 | 0.179 | 825.457 |
| three-factor model 2 | 1541.952 | 51 | 30.234 | 0.575 | 0.671 | 0.277 | 0.249 | 1368.675 |
| three-factor model 3 | 1776.177 | 51 | 34.827 | 0.508 | 0.620 | 0.298 | 0.260 | 1602.900 |
| three-factor model 4 | 1448.751 | 51 | 28.407 | 0.601 | 0.692 | 0.268 | 0.201 | 1275.474 |
| three-factor model 5 | 1651.653 | 51 | 32.385 | 0.543 | 0.647 | 0.287 | 0.202 | 1478.376 |
| three-factor model 6 | 1173.375 | 51 | 23.007 | 0.680 | 0.753 | 0.240 | 0.095 | 1000.098 |
| two-factor model 1 | 2359.961 | 53 | 44.528 | 0.367 | 0.491 | 0.338 | 0.295 | 2186.684 |
| two-factor model 2 | 2592.625 | 53 | 48.917 | 0.303 | 0.440 | 0.354 | 0.303 | 2419.348 |
| two-factor model 3 | 2967.082 | 53 | 55.983 | 0.200 | 0.357 | 0.379 | 0.309 | 2793.812 |
| two-factor model 4 | 1891.810 | 53 | 35.695 | 0.495 | 0.595 | 0.301 | 0.152 | 1718.533 |
| one-factor model | 3749.461 | 54 | 69.434 | 0.004 | 0.185 | 0.423 | 0.337 | 3576.184 |

Note: The four-factor model: the theoretical hypothesis model.

Three-factor model 1: merging team pro-social rule breaking and team innovation performance on the basis of a four-factor model.

Three-factor model 2: merging team pro-social rule breaking and team reflexivity on the basis of four-factor model.

Three-factor model 3: merging team pro-social rule breaking and team learning orientation on the basis of four-factor model.

Three-factor model 3: merging team innovation performance and team reflexivity on the basis of the four-factor model.

Three-factor model 5: merging team innovation performance and team learning orientation on the basis of four-factor model.

Three-factor Model 6: merging team learning orientation and team reflexivity on the basis of four-factor model.

Two-factor Model 1: merging team pro-social rule breaking, team reflexivity, and team innovation performance.

Two-factor model 2: merging team pro-social rule breaking, team learning orientation, and team innovation performance.

Two-factor Model 3: merging team pro-social rule breaking, team reflexivity, and team learning orientation.

Two-factor Model 4: merging team reflexivity, team learning orientation, and team innovation performance.

One-factor model: Combining all variables.

The construct validity of the variables is also examined before testing the hypotheses. A series of confirmatory factor analyses (CFAs) were conducted using Mplus8.3 to examine the distinctiveness of our study variables based on chi-square statistics and fit indices of RMSEA, CFI, and TLI. As shown in Table 3, the fit indices support that the hypothesized four-factor model of team pro-social rule breaking, team learning orientation, team reflexivity, and team innovation performance, $\chi^2 = 173.277$, df = 48; RMSEA = 0.083; CFI = 0.972; SRMR = 0.036 and TLI = 0.962, yielded a better fit to the data than the three-factor, two-factor, and one-factor models. These CFA results also provide support for the distinctiveness of the four study variables for subsequent analyses.

## Hypothesis test

Hypothesis 2 proposed that the positive relationship between team pro-social rule breaking and team reflexivity was moderated by team learning orientation. The relationships are stronger for team learning orientation is high (vs. low). The hierarchical regression results are presented in Table 4. As shown in Model 4 of Table 4, after we controlled for the control variable, we found that ethical leadership was significantly and positively related to interaction terms ($\beta = 0.302$, p <0.05). The simple slope analysis showed that when team members perceived a high level of team learning orientation (+SD), the effect of team pro-social rule breaking on team reflexivity was significant ($\beta = 0.436$, 95%CI [0.282, 0.726], excluding 0). When team members perceived low levels of team learning orientation (-SD), team pro-social rule breaking had no significant effect on team reflexivity ($\beta = -0.113$, 95% CI [−0.416, 0.117], including 0). Fig 2 and a separate

**Table 4. Hierarchical regression analysis results.**

| | TR | | | | TIP | |
|---|---|---|---|---|---|---|
| | **Model 1** | **Model 2** | **Model 3** | **Model 4** | **Model 5** | **Model 6** |
| **Control variables** | -0.139 | -0.148 | -0.043 | 0.000 | -0.096 | -0.037 |
| Edu | 0.022 | 0.024 | 0.010 | 0.012 | -0.004 | -0.013 |
| Incoming | -0.024 | -0.018 | -0.009 | -0.015 | -0.038 | -0.028 |
| Age | 0.000 | 0.001 | 0.001 | 0.002 | 0.003 | 0.003 |
| Tenure | 0.141 | 0.126 | 0.071 | 0.102 | 0.131 | 0.070 |
| Size | 0.122 | 0.117 | 0.047 | 0.025 | 0.229 | 0.176 |
| Percentage of females | -0.139 | -0.148 | -0.043 | 0.000 | -0.096 | -0.037 |
| **Independent variable** | | | | | | |
| TPSRB | | 0.176 | 0.188 | 0.162 | | |
| **Moderating variable** | | | | | | |
| TLO | | | 0.361** | 0.377** | | |
| **Interaction** | | | | | | |
| TPSRB*TLO | | | | 0.302* | | |
| **Mediating variable** | | | | | | |
| TR | | | | | | 0.429* |
| R² | 0.085 | 0.109 | 0.432** | 0.472* | 0.056 | 0.129* |
| △R² | | 0.024 | 0.323** | 0.040* | | 0.073* |
| △F | 1.163 | 1.985 | 41.485** | 5.535* | 0.743 | 6.220* |

Note: * $p < 0.05$

** $p < 0.01$.

Abbreviations: TPSRB, team pro-social rule breaking; TLO, team learning orientation; TR, team reflexivity; TIP, team innovation performance.

simple slope analysis further reveal that the positive relationship between team pro-social rule breaking and team reflexivity is stronger for teams with high levels of learning orientation. Thus, hypothesis 2 was supported. See Table 3 for hierarchical regression analysis results.

Hypothesis 3 proposed that team reflexivity was positively related to team innovation performance. As shown in Model 6 of Table 4, after we controlled for the control variable, we found that team reflexivity was significantly and positively related to team innovation performance ($\beta = 0.429$, p <0.05). Thus, Hypothesis 3 was supported.

As Table 5 shows, when the level of team learning orientation is high, the relationship between team pro-social rule breaking and team innovation performance is significant and positive ($\beta = 0.187$, 95% CI [0.038, 0.348], excluding 0). Correspondingly, when the level of team learning orientation is low, the relationship between team pro-social rule breaking and team innovation performance through team reflexivity is negative, but the relationship is not significant ($\beta = -0.048$, 95% CI [−0.230, 0.037], including 0). The results indicate that the low level of team learning orientation cannot effectively stimulate team reflexivity. One possible explanation is that the low level of team learning orientation leads team members to ignore the behavioral characteristics of team pro-social rule breaking, which is consistent with the previous research results on the impact of team learning orientation on organizational innovation performance [64]. The difference between the two levels is significant, with 95% CI [0.039, 0.559], excluding 0. Therefore, hypothesis 4 was supported.

## Discussion

Team cooperation has progressively become the fundamental work pattern across various organizations, and organizational research, aligning with this industry trend, has shifted its

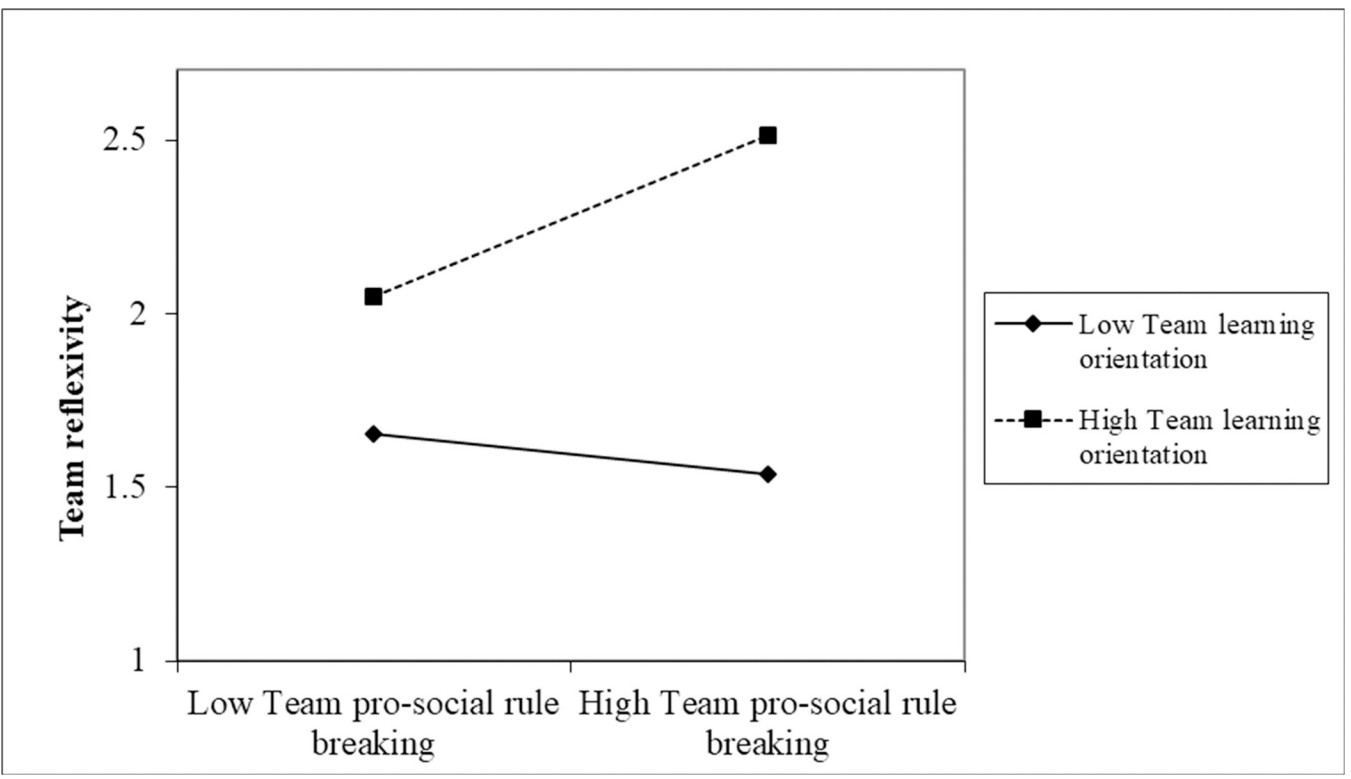

**Fig 2. The interaction between team pro-social rule breaking and team learning orientation on team reflexivity.**

focus toward understanding team behavior [10]. Following the operational methods of previous studies [51], this research used the mean value of team members' pro-social rule breaking to represent the team's pro-social rule breaking. Drawing from social information processing theory, a moderated mediation model was constructed to elucidate the relationship between team pro-social rule breaking and team innovation performance. The hypothesis was tested using data collected from 382 participants across two companies. The results demonstrated that team learning orientation moderated the relationship between team pro-social rule breaking and team reflexivity when team learning orientation is high, and team reflexivity mediated the relationship between the interaction (team pro-social rule breaking and team learning orientation) and team innovation performance. Specifically, compared with the lower team learning orientation, the interaction effect on team innovation performance through team reflexivity is stronger when the team learning orientation is higher. The result also revealed that team pro-social rule breaking does not directly impact team reflexivity and team innovative performance. Only in an climate of pursuing learning can team pro-social rule-breaking behavior influence team reflexivity.

**Table 5. Results of moderated mediation effect.**

| Subgroup Statistics | Coefficient | S.E. | 95% confidence interval | |
|---|---|---|---|---|
| | | | Low | High |
| Conditional indirect effects | | | | |
| team learning orientation (+1 SD) | 0.187 | 0.077 | 0.038 | 0.348 |
| team learning orientation (-1 SD) | -0.048 | 0.064 | -0.230 | 0.037 |
| Diff | 0.236 | 0.126 | 0.039 | 0.559 |

The results of this study reveal an important phenomenon: Team pro-social rule breaking can stimulate individuals to play more, actively thinking and enhance their creative effort in teams with a high-level learning orientation. However, a noteworthy implication of our analysis is that certain behaviors may not trigger team members' reflective thinking in teams with a low-level learning orientation. Learning orientation is a crucial team condition that dictates individuals' importance on team behavior and subsequent responses. These research results align with the findings of Hirst (2009) and other scholars on team learning orientation. A learning-oriented team style makes individuals more attuned to team behavior trends, prompting them to adjust their behaviors accordingly [43]. Additionally, in the Chinese context, organizations often emphasize employee obedience and conformity [81]. In teams with low learning tendencies, pro-social rule breaking behaviors are more likely to be regarded as disrespectful behaviors to organizational rules. This can further incite negative feedback from employees regarding such behaviors without necessarily prompting reflection on the existing work mode.

## Theoretical implications

Our study makes several contributions to the literature. First, we identify the types of pro-social rule breaking at the team level. Previous studies focused more on the antecedents and consequences of individual pro-social rule breaking but neglected the study of team pro-social rule breaking [4, 5, 7, 8, 15, 35]. Team pro-social rule breaking is a unique group phenomenon that refers to the prevalence of work team members engaging in pro-social rule breaking. This feature of team participation makes the impact of team pro-social rule breaking on the organization more significant than that of individual behaviors. This study provides a theoretical basis for defining pro-social rule breaking as a group-level phenomena, opens the "black box" of team pro-social rule breaking, and affords a theoretical reference for future research. In addition, the result also responds to the problem of "solving whether positive deviant behaviors are contagious" in the field of positive deviant behaviors by constructing the concept of pro-social rule breaking at the team level [82]. That is, pro-social rule breaking in the team is contagious. The behavior is not just carried out by a single employee but an conduct that can spread. This study also revealed the positive effect of pro-social rule breaking on team innovation performance. Different from previous studies that focused on the negative results of pro-social rule breaking [5, 9], the results of this study demonstrated the promotion of pro-social rule breaking on organizational innovation ability and expanded the positive consequences of pro-social rule breaking.

Second, we extend the antecedent study of team reflexivity by exploring the impact of team pro-social rule breaking on team reflexivity. Previous studies focused more on the influence of organizational factors and individual traits on team reflexivity [2–8] but ignored the influence of team factors on team reflexivity. By exploring the relationship between team pro-social rule breaking and team reflexivity, we found that team pro-social rule breaking positively affects team innovation performance through team reflexivity. In summary, by exploring the mediating role of team reflexivity in the influence of team pro-social rule breaking on team innovation performance, this study expands the research on the anthems of team reflexivity and promotes the formation of a "pre-outcome" closed loop in the study of pro-social rule breaking.

Third, the research results show that team learning orientation, as an organizational boundary condition, can trigger team members' reflexivity on team behaviors. This research result responds to scholars' call for research on team learning orientation [83] and strengthens the influence of interaction between team learning orientation and organizational factors on team

output. As an important team characteristic, team learning orientation inspires team members to attach importance to team pro-social rule breaking and objectively contributes to team reflexivity. The results of this study clarified the boundary conditions of team pro-social rule breaking affecting team reflexivity. Furthermore, they explained the differences in team pro-social rule breaking affecting team innovation performance under various team characteristics.

## Practical implications

Overall, this study provides insight into organizational management and human resource practices.

First, the research shows that pro-social rule breaking in the organization can become a positive team phenomenon, therefore managers should be aware. Considering the widespread application of group work patterns in today's organizations, the effect of team pro-social rule breaking on the organization is more significant than the effect of employee behaviors, thus managers must value it highly. Managers should understand the true purpose of team violations and consider the complex impact of the diversity of organizational behavior on individuals and other stakeholders. Only in this way can the team be appropriately managed and guided, and the organization rules should be adjusted in time to adapt to the rapid changes in the environment. As not all employees' violations aim to harm the organization or seek personal benefits, we should make full use of their goodwill. Although this study shows that we have no reason to support employees to carry out more pro-social rule breaking, our study also emphasizes that managers should not blindly punish employees who break the rules, dampening their enthusiasm and even resulting in the counter-productive work behavior of the whole team.

Second, managers should understand the antecedence mechanism of team reflexivity. Through effective stimulation and guidance of team reflexivity, employees can constantly improve themselves in reflexivity and ultimately promote organizational performance improvement. This study shows that pro-social rule breaking is an opportunity for team reflexivity. If team members can receive information or feedback from others through communication and discussion, they are more likely to generate creative ideas. Therefore, managers should build team communication channels to develop an environment where team members are encouraged to share their ideas, openly discuss their experiences, and learn about each other's tasks, goals, and progress through frequent discussions and collaboration. Therefore, creating an atmosphere of communication within the team and effectively stimulating team reflexivity are powerful measures to improve team innovation performance.

Finally, the conditional effect of team learning orientation reminds managers that it is necessary to establish learning values and create a positive learning atmosphere. Managers should establish learning-oriented values in the organization and an organizational culture that prioritizes learning within the team. These measures can help the team take full advantage of pro-social rule breaking and promote team reflexivity, thus improving team innovation performance.

## Limitations and future research

There are several limitations in this study. First, the data collection method in this paper has limitations. The research data were collected by questionnaire, and the collection method was relatively simple. In terms of research methods, there are deficiencies in research design. This study uses three different time points to collect the data on team pro-social rule breaking, team reflexivity, and team innovation performance to reduce the impact of common method

deviation [57]. However, the pro-social rule breaking of the team rises to the team level by taking the average value, and the problem of the validity of concept measurement may still exist. Future research can assist case and qualitative research.

Second, there are limitations in the survey objects of this study. The survey objects of our study are general manufacturing employees from Jiangsu, Zhejiang, and Shanghai regions, and the sample characteristics do not reflect the regional or industry diversity. Although the research results have important guiding significance for the management practice under the Chinese background, it also brings about the external effectiveness of the results and cross-cultural promotion.

Third, the impact of team pro-social rule breaking results is not detailed and in-depth. Morrison (2006) divided pro-social rule breaking into three different dimensions: to improve work efficiency, help colleagues, and provide better services to customers [2]. Employees' behaviors of violating rules for different pro-social motives may have different results on employees' own performance or team performance. For example, Mayer et al. studied in more detail the customer-oriented employees' pro-social rule breaking [84]. This study did not explore the impact of different types of team pro-social rule breaking on team performance. The follow-up study can examine the impact mechanism of different team pro-social rule breaking dimensions on the outcome variables.

## Conclusion

Drawing upon information processing theory, we develop a moderated mediation model to examine the relationship between team pro-social rule breaking and team innovation performance. Firstly, referring to the research of other team concepts, our study defines the concept of team pro-social rule breaking and the operational method. Then, 382 sample data from two branches of a large manufacturing enterprise are used to test the model. The analysis results confirm all the hypotheses proposed in our study. Finally, this study discusses the theoretical and managerial contributions of the study.

In addition, our research provides a theoretical basis for management to enhance team creativity. Managers often need to manage teams in complex organizational environments to maintain team innovation efforts. Our research provides feasible solutions to solve this problem, and confirms the positive effect of the combination of team learning orientation and team pro-social rule breaking on team innovation. The management should be more rational about the positive significance of the team's pro-social rule breaking to the organization, and should enhance the organization's learning initiative and increase employees' self-reflection through active efforts.

## Author Contributions

**Conceptualization:** Guosen Miao, Guoping Chen.

**Data curation:** Guosen Miao, Ying Yao.

**Formal analysis:** Guosen Miao.

**Investigation:** Guosen Miao, Ying Yao.

**Methodology:** Guosen Miao, Guoping Chen.

**Resources:** Guosen Miao, Guoping Chen.

**Writing – original draft:** Guosen Miao.

**Writing – review & editing:** Guosen Miao, Guoping Chen.

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
