## [Decision Letter · Decision Letter 0]

29 Aug 2023

PONE-D-23-17868Team pro-social rule breaking and team innovation performance: An information processing theory perspectivePLOS ONE

Dear Dr. Miao,

Thank you for submitting your manuscript to PLOS ONE. After careful consideration, we feel that it has merit but does not fully meet PLOS ONE’s publication criteria as it currently stands. Therefore, we invite you to submit a revised version of the manuscript that addresses the points raised during the review process.

The reviewers have raised several major issues with the paper regarding theory, hypotheses development, methodology, etc. I agree with many of the reviewer comments. A substantial amount of effort is expected from the authors in revising this paper and addressing all of the reviewer comments.

We look forward to receiving your revised manuscript.

Kind regards,

Sharjeel Saleem

Academic Editor

PLOS ONE

2. Please upload a copy of Figure 2, to which you refer in your text on page 13. If the figure is no longer to be included as part of the submission please remove all reference to it within the text.

Reviewers' comments:

Reviewer's Responses to Questions

**Comments to the Author**

1. Is the manuscript technically sound, and do the data support the conclusions?

Reviewer #1: No

Reviewer #2: Yes

Reviewer #3: Partly

2. Has the statistical analysis been performed appropriately and rigorously? 

Reviewer #1: Yes

Reviewer #2: Yes

Reviewer #3: No

3. Have the authors made all data underlying the findings in their manuscript fully available?

Reviewer #1: Yes

Reviewer #2: Yes

Reviewer #3: No

4. Is the manuscript presented in an intelligible fashion and written in standard English?

Reviewer #1: No

Reviewer #2: Yes

Reviewer #3: No

5. Review Comments to the Author

Reviewer #1: The objective of the paper " Team pro-social rule breaking and team innovation performance:An information processing theory perspective" is interesting but the current version of manuscript needs a radical new structure and development.

First, the aim should be evidenced in the abstract and introduction sections.

Second, the methodological approach should be explained in a clearer way.

Third, the contribution can be made evident only putting the accent on the gap in the literature.

Finally, the interpretation of results are not consistent with the objectives and need further improvement also in terms of policy implications.

Reviewer #2: Authors have done wonderful work appreciation to the authors. There few concerns which authors need to incorporate

1. Add Factor loadings table

2. Add reliability of each construct/variable

3. Add Average variance extracted and composite reliability

4. Discriminant validity is missing (Fornell Larcker criteria or HTMT ratios can be sufficient)

5. Enhance discussion section and compare hypotheses with past studies

6. Add meaningful information in cocnlusion section

Reviewer #3: • The central premise of the paper is built on the relationship between team prosocial rule breaking with innovation performance which hasn’t been checked before however the relationship between team reflexivity and innovation has already been established by extant research that limits the theoretical contributions and novelty of this article. Similarly extant research has studied and found positive relationship between team learning goal orientation and team reflexivity.

• The relationship between Team prosocial rule breaking and Team reflexivity has not been studied before and auhor has proposed the moderating effect of learning orientation without establiblishing the prime hypothesis between Team prosocial rule breaking and Team reflexivity. I suggested here to first establish and justify the relationship between these two variables with soild theoretical argumentation in separate hypothesis and than check the moderation effect.

• At numerous places author had claimed blunt statemt without proper refernance. E.g “Ac-cording to the social information processing theory, as an individual capable of self-regulation, an employee's subsequent behavior not only depends on the past behavior, but also is influenced by environmental cues.(ref ???) Following this logic, individuals' subsequent responses to collective team violations can be influenced by team environmental cues such as team learning orientation.(ref???) Team learning orientation is a series of values that influence the creation and utilization of knowledge by a team and shape the learning atmosphere of a team.(ref???)” requires strong literature support.

• This study has defined multiple ideas in one frame without solid justification and theoretical understanding. The autor need to rethink the ideas and frame them in a structure way with solid argumentaion.

• This study is backed by numerous theories and multiple conflicting ideas. The author may highlight the role of overarching theory to explain the proposed relationships in all hypothesis..

• Clarity is required regarding the positioning of the article and its theoretical contributions. All hypothesized relationship needs proper references and theoratica argumentation like in H2.

• The paper need restructuring of ideas and must be in acceptable language as I have seen numerous grammatical and typo mistakes. E.g

• Line 42: “…workplace when they in-novate”- the mistake of putting a hyphen in a word has been made several times throughout the paper.

• Line 43: “…subjects of innovation activity”- drivers is innovation activity is a more appropriate phrase.

• Line 43: “…employees inevitably challenge and break with practices”- It should be break away.

• Line 63: “…and this process that is known as team reflexivity”-this is a grammatically incorrect or an incomplete sentence.

• Line 68: “In other words, team members accumulate and evaluate the information conveyed by the contradictions in past actions by observing themselves or the environment in order to make adaptations and changes to the environment”. This sentence is very ambiguous and does not make the purpose of writing the sentence clear to the reader.

• Line 86: “The theoretical model framework of this study was depicted in Figure 1 as follows”- this is a grammatically incorrect sentence.

• Line 88: “First, we attempt to identify the types of pro-social rule breaking at the team level fill a notable perspective gap.” This is a grammatically incorrect sentence.

• Line 89: “Previous research has focused merely on pro social rule breaking at the individual lev-el and has not yet focused on the team level.” This sentence requires citations.

• Line 102: Previous research has focused merely on pro social rule breaking at the individual lev-el and has not yet focused on the team level. This is grammatically incorrect sentence.

• Line 104, 105, 106, 107: Firstly, team members are a crucial factor of work environment, team members are able to inform certain activities and help others shape beliefs about what should do or not do [9, 25]. People can signal from the fact that colleagues act pro-social rule breaking behavior for the benefit of the organization or stakeholders.

• The above sentences are grammatically incorrect and ambiguous, failing to make any meanings.

• Line 109-110: Secondly, pro-social rule breaking refers to help organizations, colleagues, or customers with altruistic intentions, even if leading to negative results. This sentence is grammatically incorrect.

• Line 111: “…and then likely to imitate in the future.” This is grammatically incorrect.

• I have serious concerned with the methodological section of this paper as the validity of construct has not reported in paper. Author has just reported main variable context resuts and not compare every variable with other in factors model. Bootstrap sample size is missing , similarly I have not found ICCI,ICC2 or RWG test. The author also has not reported moderation graph. Apart from that I also suggest to report the values of simple slope analysis in interpretation with justification that what happened at high and low levels.

---

## [Author Response · Author response to Decision Letter 0]

19 Oct 2023

Reviewer #1: The objective of the paper " Team pro-social rule breaking and team innovation performance: An information processing theory perspective" is interesting but the current version of manuscript needs a radical new structure and development.

First, the aim should be evidenced in the abstract and introduction sections.

Our response: Thanks very much for your helpful suggestions. We have optimized the abstract and introduction sections to highlight the main purpose of this paper. Please refer to the respective sections in the revised manuscript for details (Line 30-31, 43-45, 59-61, 65-67).

Second, the methodological approach should be explained in a clearer way.

Our response: Thanks very much for your helpful suggestions. We have added a description of the data aggregation process for TPSRB in the methods section, along with explanations of variable reliability and validity testing. To ensure clarity in describing the methodology, we have included an overview of the method's implementation process in the methods and data analysis sections. Please refer to the specific sections for detailed information (Line 361-370, 382-392,).

Third, the contribution can be made evident only putting the accent on the gap in the literature.

Our response: Thanks very much for your helpful suggestions. We acknowledge that the limitations in the paper's contributions stemmed from not adequately emphasizing the gaps in the literature. To address this issue, members of the research team reevaluated relevant literature and engaged in focused discussions on PSRB, team reflexivity and team learning oriented papers. Based on a review of past literature, we identified the main gaps in the following areas: 1. Research on PSRB primarily focuses on the individual level, with results mostly revealing the negative effects of PSRB on individual performance evaluations. There is a lack of research on PSRB at the team level and exploration of its positive effects. 2. Studies on factors influencing team reflexivity have concentrated more on organizational contextual factors and team member traits, with a shortage of investigations into how team behavioral tendencies impact the team reflexivity mechanism. 3. Research on team learning orientation has predominantly focused on its direct effects on organizational outcomes, with limited studies on the interaction effects of team learning orientation with other organizational characteristics in complex organizational contexts. This paper reexamines these identified gaps and builds upon them to formulate the paper's contributions. For specific details, please refer to the Introduction section (line 79 to 97).

Finally, the interpretation of results are not consistent with the objectives and need further improvement also in terms of policy implications.

Our response: Thanks very much for your helpful suggestions. We conducted a reanalysis of the data analysis results and summarized the support for our hypotheses. According to the results of hierarchical regression analysis, the interaction between team learning orientation and TPSRB has a positive direct effect on team reflexivity, consistent with Hypothesis 1. Additionally, the results support the positive role of team reflexivity in team innovation performance, aligning with Hypothesis 2. Finally, we examined the moderated mediation effect, and the results indicated that only under high levels of team learning orientation, TPSRB's mediated effect on team innovation performance through team reflexivity is established, in line with Hypothesis 3. Furthermore, based on the findings of this study, we have updated the managerial implications, emphasizing how managers in complex organizational contexts can enhance employee innovation performance by comprehensively understanding and promoting TPSRB. For specific details, please refer to the specific sections (Line 522-524).

Reviewer #2: Authors have done wonderful work appreciation to the authors. There few concerns which authors need to incorporate

1. Add Factor loadings table

Our response: Thanks very much for your helpful suggestions. Your suggestions have helped us improve the rigor of our data analysis. We have added a factor loading table, CR and AVE in the data analysis chapter. You can review the details in the data analysis chapter (line 361 to 371).

2. Add reliability of each construct/variable

Our response: Thanks very much for your helpful suggestions. The Cronbach's alpha for all constructs is between 0.919 and 0.984, which has been reported in the variable Measure chapter. Moreover, the composite reliability (CR) of variables is between 0.921 and 0.984, which has also been reported in the variable Measure chapter and is in line with the recommendations of scholars. Based on the above analysis results, we believe that the reliability of the measurement results is acceptable.

3. Add Average variance extracted and composite reliability

Our response: Thanks very much for your helpful suggestions. We have added a description of the CR and AVE results in the data analysis chapter. You can also find specific numerical values in the table above.

4. Discriminant validity is missing (Fornell Larcker criteria or HTMT ratios can be sufficient)

Our response: Thanks very much for your helpful suggestions. We used Fornell-Larcker criteria to test the discriminative validity of variables, as follows:

We employed the Fornell-Larcker criterion to assess construct discriminant validity. The square root of AVEs (Average Variance Extracted) ranged from 0.702 to 0.922, while correlation coefficients between the variables ranged from 0.004 to 0.615. In all cases, the square root of AVEs exceeded the correlation coefficients between variables, providing confirmation of discriminant validity (Line 389-392).

5. Enhance discussion section and compare hypotheses with past studies

Our response: Thank you for your suggestions, and we apologize for the insufficiency in the content of the discussion section. We have restructured the discussion section, including a comparison of the data analysis results with existing research. We have reaffirmed the positive impact of the high-level team learning orientation and TPSRB interaction on team innovation performance, which aligns with previous research (Hirst, 2009). Additionally, we have provided an explanation for the non-establishment of the mediated effect under the premise of low-level team learning orientation. The specific content is as follows:

The results of this study reveal an important phenomenon that team pro-social rule breaking can stimulate individuals to play more active thinking and enhance their creative effort in teams with high level learning orientation. However, a noteworthy implication of our analysis is that in teams with low level learning orientation, certain behaviors may not trigger team members' reflective thinking. Learning orientation, serving as a crucial team condition, dictates the importance individuals place on team behavior and their subsequent responses. These research results align with the findings of Hirst (2009) and other scholars on team learning orientation. A learning-oriented team style makes individuals more attuned to team behavior trends, prompting them to adjust their behaviors accordingly (Hirst, 2009). Additionally, in the Chinese context, organizations often emphasize employee obedience and conformity (Farh, 2000), and in teams with low learning tendency, pro-social rule breaking behaviors are more likely to be regarded as disrespectful behaviors to organizational rules. This can further incite negative feedback from employees regarding such behaviors, without necessarily prompting reflection on the existing work mode.

Hirst G, Van Knippenberg D, Zhou J. A cross-level perspective on employee creativity: Goal orientation, team learning behavior, and individual creativity[J]. Acad manage j, 2009, 52(2): 280-293.

Farh J L, Cheng B S. A cultural analysis of paternalistic leadership in Chinese organizations[M]//Management and organizations in the Chinese context. London: Palgrave Macmillan UK, 2000: 84-127.

6. Add meaningful information in conclusion section

Our response: Thank you for your suggestions. We added the discussion of the research results in the conclusion chapter, and on the basis of summarizing the research results, we emphasized the positive effect of the research on improving team creativity. At the same time, it also puts forward management practice measures based on the research results, the specific contents are as follows:

Our research provides a theoretical basis for management to enhance team creativity. Managers often need to manage teams in complex organizational environments to maintain team innovation efforts. Our research provides feasible solutions to solve this problem, and confirms the promotion effect of the combination of team learning orientation and team pro-social rule breaking on team innovation. The management should be more rational about the positive significance of the team's pro-social rule breaking to the organization, and should enhance the organization's learning initiative and increase employees' self-reflection through active efforts.

Reviewer #3: • The central premise of the paper is built on the relationship between team prosocial rule breaking with innovation performance which hasn’t been checked before however the relationship between team reflexivity and innovation has already been established by extant research that limits the theoretical contributions and novelty of this article. Similarly extant research has studied and found positive relationship between team learning goal orientation and team reflexivity.

Our response: Thank you for your suggestions. Your suggestions have been helpful in improving the structure of our article, clarifying the shortcomings of existing research, and summarizing the theoretical contributions of this paper. Following your advice, we have reorganized the relevant literature on team reflexivity and team innovation performance, as well as summarized the limitations of related research. On the one hand, although some studies have indicated the positive impact of team reflexivity on organizational innovation performance (Fu, 2021), research on the factors influencing team reflexivity and subsequently promoting organizational innovation performance remains limited to organizational factors and the influence of team member traits, lacking a more diverse range of research perspectives. Based on the theory of social information processing, individuals will adjust their behavior within the organizational environment in which they study (Salancik, 1978). The premise of this adjustment involves the analysis of collected information (Chen, 2013). Team members cannot automatically process information generated by ambiguous behavior of pro-social rule breaking, instead, they use a systematic information processing approach that triggers deeper thinking among team members about whether there are issues with the behavior or the rules. This, in turn, leads to a shared perception within the team. Consequently, team members are capable of critically assessing and reflecting on whether the team's behavior is suitable for the current or intended environment, a process known as team reflexivity. Team reflexivity refers to the process by which team members openly reflect on team goals, strategies, and overall business processes to adapt to changes in the environment (Schippers, 2008). Previous studies have indicated that team reflexivity can encourage employees to develop an understanding of the organization's work goals by observing organizational factors such as the human resource system, leadership styles, and the diversity of team goals, ultimately leading to more innovative behaviors (Wang, 2020; Chen, 2016; Nederveen, 2011; Dayan, 2010). Although research on the influence of team reflexivity on creativity performance has unveiled the mechanism of team reflexivity on individual innovation investment, research on the external factors that trigger team reflexivity has primarily focused on the organizational and individual levels and has not explored the impact of team situational factors on team reflexivity (Hoegl, 2006). As mentioned above, team members interact more frequently during working hours (Ren, 2021; Kozlowski, 2012) and have more consistent work goals and clear work divisions (Cooper, 2019). Consistent behavior patterns within a team, as an important source of organizational information, may be more stimulating for individuals to reflect on work goals than other situational factors. Importantly, this reflective process involves the analysis of the current organizational situation and teamwork. In comparison to well-known psychological processes such as psychological contract breach (Kim, 2018) and leadership identification (DeRue, 2010), this cognitive process can better emphasize the impact of social information processing on employees’ cognition and behavior. Therefore, from the perspective of social information processing theory, this study posits that team factors, such as team pro-social rule-breaking, may stimulate team members’ reflexivity, which will affects team innovation performance further. Conditionally, team reflexivity may serve as the mediating factor linking team pro-social rule-breaking to team innovation performance.

The other hand，according to social information processing theory, the cognition of individuals within the team (e.g., team reflexivity) depends not only on past rule breaking but is also influenced by cues from the environment (Narayan, 2021). As an important team behavior orientation, team learning orientation is an environmental condition with strong information clues among various factors at the team level (Sheng, 2016). Team learning orientation represents a high level of learning commitment and open-mindedness, and encourages team members to identify, confront, and continually correct problems with existing rules or behaviors (Chiu, 2021; Atitumpong, 2018; Sinkula, 1997). Thus, when organizations emphasize learning orientation, their members are more likely to openly question the irrational rules that govern organizational behavior within the team (Alerasoul, 2021), which in turn further enhances the team reflexivity process within the team. Research on the influence of team learning orientation on creativity performance has indicated that organizational learning plays a pivotal role in enhancing organizational innovation capability and knowledge utilization (Patky, 2020; López, 2006). From a human resources perspective, learning orientation work pattern embodies a set of methodologies for disseminating and applying knowledge within an organization, allowing for the transfer of acquired knowledge into the workplace, thus creating capabilities that are both effective and efficient (Song, 2008). The research on organizational learning under various organizational scenarios has expanded scholars' insights into the interaction of team learning orientation and organizational characteristics on organizational output (Lumpkin, 2005). However, scholars' research on the interaction effects between team learning orientation and organizational characteristics in complex organizational contexts on organizational innovation output remains limited (Alerasoul, 2021). Existing studies cannot adequately explain whether an organizational atmosphere emphasizing learning can effectively stimulate individuals' deep understanding of organizational characteristics and subsequent organizational behaviors. For responding to the academic community's call for team learning-oriented research (Alerasoul, 2021; Patky, 2020; López, 2006), we introduce team learning orientation as the boundary condition. The result will extend the research boundaries related to pro-social rule-breaking and team learning orientation.

Kozlowski, S. W. J., & Bell, B. S. Work Groups and Teams in Organizatio

---

## [Decision Letter · Decision Letter 1]

11 Dec 2023

PONE-D-23-17868R1Team pro-social rule breaking and team innovation performance: An information processing theory perspectivePLOS ONE

Dear Dr. Miao,

Thank you for submitting your manuscript to PLOS ONE. After careful consideration, we feel that it has merit but does not fully meet PLOS ONE’s publication criteria as it currently stands. Therefore, we invite you to submit a revised version of the manuscript that addresses the points raised during the review process. Although some comments have been addressed in your revision. However, reviewer 3 has still raised serious concerns regarding the crafting of the paper. As recommended by reviewer 3, you are required to not only mention your revisions in the response letter but also you have to make corresponding changes in the revised manuscript. Please address all of the reviewer 3 comments adequately and deeply, and make the appropriate changes in the revised manuscript. The changes made in the revised manuscript should be highlighted for ease of the reviewer.   Please submit your revised manuscript by Jan 25 2024 11:59PM. If you will need more time than this to complete your revisions, please reply to this message or contact the journal office at plosone@plos.org. Please include the following items when submitting your revised manuscript:A rebuttal letter that responds to each point raised by the academic editor and reviewer(s). You should upload this letter as a separate file labeled 'Response to Reviewers'.A marked-up copy of your manuscript that highlights changes made to the original version. You should upload this as a separate file labeled 'Revised Manuscript with Track Changes'.An unmarked version of your revised paper without tracked changes. You should upload this as a separate file labeled 'Manuscript'.

We look forward to receiving your revised manuscript.

Kind regards,

Sharjeel Saleem

Academic Editor

PLOS ONE

Reviewers' comments:

Reviewer's Responses to Questions

**Comments to the Author**

1. If the authors have adequately addressed your comments raised in a previous round of review and you feel that this manuscript is now acceptable for publication, you may indicate that here to bypass the “Comments to the Author” section, enter your conflict of interest statement in the “Confidential to Editor” section, and submit your "Accept" recommendation.

Reviewer #2: All comments have been addressed

Reviewer #3: (No Response)

2. Is the manuscript technically sound, and do the data support the conclusions?

Reviewer #2: Yes

Reviewer #3: No

3. Has the statistical analysis been performed appropriately and rigorously? 

Reviewer #2: Yes

Reviewer #3: Yes

4. Have the authors made all data underlying the findings in their manuscript fully available?

Reviewer #2: Yes

Reviewer #3: Yes

5. Is the manuscript presented in an intelligible fashion and written in standard English?

Reviewer #2: Yes

Reviewer #3: No

6. Review Comments to the Author

Reviewer #2: Authors have incorporated the comments and answered to all the queries of the reviewers. Therefore In my opinion it is recommended for publication in PONE journal.

Reviewer #3: I feel great difficulty to understand the revision as the author has not highlighted the portion which they have revised, secondly I found the grammatical issues along with lack of theoretical justification in revision , however they have made proper justification in remarks portion. I suggest to incorporate these changes in manuscript along with highlighting the revision portion.

---

## [Author Response · Author response to Decision Letter 1]

22 Jan 2024

Thank you reviewer 3 and editor comments for helpful guidance on the completeness and scientificity of the manuscript. In order to further answer your questions, we have reorganized the last round of replies to make the content easier to understand. Meanwhile, the research team has highlighted the revisions to the manuscript, hoping that our efforts on the manuscript can help reviewers better understand the revised content. Below is our response to your query and a highlighted explanation of the changes in the manuscript.

Reviewer #3: The central premise of the paper is built on the relationship between team prosocial rule breaking with innovation performance which hasn’t been checked before however the relationship between team reflexivity and innovation has already been established by extant research that limits the theoretical contributions and novelty of this article. Similarly extant research has studied and found positive relationship between team learning goal orientation and team reflexivity.

Our response: Thank you for your suggestions. Your suggestions have been helpful in improving the structure of our article, clarifying the shortcomings of existing research, and summarizing the theoretical contributions of this paper. Following your advice, we have reorganized the relevant literature on team reflexivity and team innovation performance, as well as summarized the limitations of related research. On the one hand, although some studies have indicated the positive impact of team reflexivity on organizational innovation performance (Fu, 2021), research on the factors influencing team reflexivity and subsequently promoting organizational innovation performance remains limited to organizational factors and the influence of team member traits, lacking a more diverse range of research perspectives. Based on the theory of social information processing, individuals will adjust their behavior within the organizational environment in which they study (Salancik, 1978). The premise of this adjustment involves the analysis of collected information (Chen, 2013). Team members cannot automatically process information generated by ambiguous behavior of pro-social rule breaking, instead, they use a systematic information processing approach that triggers deeper thinking among team members about whether there are issues with the behavior or the rules. This, in turn, leads to a shared perception within the team. Consequently, team members are capable of critically assessing and reflecting on whether the team's behavior is suitable for the current or intended environment, a process known as team reflexivity. Team reflexivity refers to the process by which team members openly reflect on team goals, strategies, and overall business processes to adapt to changes in the environment (Schippers, 2008). Previous studies have indicated that team reflexivity can encourage employees to develop an understanding of the organization's work goals by observing organizational factors such as the human resource system, leadership styles, and the diversity of team goals, ultimately leading to more innovative behaviors (Wang, 2020; Chen, 2016; Nederveen, 2011; Dayan, 2010). Although research on the influence of team reflexivity on creativity performance has unveiled the mechanism of team reflexivity on individual innovation investment, research on the external factors that trigger team reflexivity has primarily focused on the organizational and individual levels and has not explored the impact of team situational factors on team reflexivity (Hoegl, 2006). As mentioned above, team members interact more frequently during working hours (Ren, 2021; Kozlowski, 2012) and have more consistent work goals and clear work divisions (Cooper, 2019). Consistent behavior patterns within a team, as an important source of organizational information, may be more stimulating for individuals to reflect on work goals than other situational factors. Importantly, this reflective process involves the analysis of the current organizational situation and teamwork. In comparison to well-known psychological processes such as psychological contract breach (Kim, 2018) and leadership identification (DeRue, 2010), this cognitive process can better emphasize the impact of social information processing on employees’ cognition and behavior. Therefore, from the perspective of social information processing theory, this study posits that team factors, such as team pro-social rule-breaking, may stimulate team members’ reflexivity, which will affects team innovation performance further. Conditionally, team reflexivity may serve as the mediating factor linking team pro-social rule-breaking to team innovation performance.

The other hand，according to social information processing theory, the cognition of individuals within the team (e.g., team reflexivity) depends not only on past rule breaking but is also influenced by cues from the environment (Narayan, 2021). As an important team behavior orientation, team learning orientation is an environmental condition with strong information clues among various factors at the team level (Sheng, 2016). Team learning orientation represents a high level of learning commitment and open-mindedness, and encourages team members to identify, confront, and continually correct problems with existing rules or behaviors (Chiu, 2021; Atitumpong, 2018; Sinkula, 1997). Thus, when organizations emphasize learning orientation, their members are more likely to openly question the irrational rules that govern organizational behavior within the team (Alerasoul, 2021), which in turn further enhances the team reflexivity process within the team. Research on the influence of team learning orientation on creativity performance has indicated that organizational learning plays a pivotal role in enhancing organizational innovation capability and knowledge utilization (Patky, 2020; López, 2006). From a human resources perspective, learning orientation work pattern embodies a set of methodologies for disseminating and applying knowledge within an organization, allowing for the transfer of acquired knowledge into the workplace, thus creating capabilities that are both effective and efficient (Song, 2008). The research on organizational learning under various organizational scenarios has expanded scholars' insights into the interaction of team learning orientation and organizational characteristics on organizational output (Lumpkin, 2005). However, scholars' research on the interaction effects between team learning orientation and organizational characteristics in complex organizational contexts on organizational innovation output remains limited (Alerasoul, 2021). Existing studies cannot adequately explain whether an organizational atmosphere emphasizing learning can effectively stimulate individuals' deep understanding of organizational characteristics and subsequent organizational behaviors. For responding to the academic community's call for team learning-oriented research (Alerasoul, 2021; Patky, 2020; López, 2006), we introduce team learning orientation as the boundary condition. The result will extend the research boundaries related to pro-social rule-breaking and team learning orientation.

Based on the above contents, we have revised the relevant contents of the introduction chapter. For details, please refer to the contents highlighted in yellow (lines 80-83, 112-116, 119-137).

Reference

Kozlowski, S. W. J., & Bell, B. S. Work Groups and Teams in Organizations. Handbook of Psychology, Second Edition. Wiley: New York, 2012, pp. 412-469.

Ren, S., Wang, Z., & Collins, N. T. The joint impact of servant leadership and team-based HRM practices on team expediency: the mediating role of team reflexivity. Pers Rev 2021, Vol.50: 1757-1773. doi:10.1108/PR-07-2020-0506.

Salancik, G. R., & Pfeffer, J. A Social Information Processing Approach to Job Attitudes and Task Design. Admin Sci Quart 1978, 23(2), 224-253. doi:10.2307/2392563.

Chen, Z., Takeuchi, R., & Shum, C. A social information processing perspective of coworker influence on a focal employee. Organ Sci 2013, 24(6), 1618-1639. doi: 10.1287/orsc.2013.0820.

Cooper, B., Wang, J., Bartram, T., & Cooke, F. L. Well‐being‐oriented human resource management practices and employee performance in the Chinese banking sector: The role of social climate and resilience. Hum Resour Manage-Us 2019, 58(1), 85-97.

Schippers, M. C., Den Hartog, D. N., Koopman, P. L., & van Knippenberg, D. The role of transformational leadership in enhancing team reflexivity. Hum Relat 2008, 61(11), 1593-1616. doi:10.1177/0018726708096639.

Chen, S., Zhang, G., Zhang, A., & Xu, J. Collectivism-oriented human resource management and innovation performance: An examination of team reflexivity and team psychological safety. J Manage Organ 2016, 22(4), 535. doi:10.1017/jmo.2015.50.

Wang, Z., Ren, S., Chadee, D., Liu, M., & Cai, S. Team reflexivity and employee innovative behavior: the mediating role of knowledge sharing and moderating role of leadership. J Knowl Manag 2020, 25(6): 1619-1639. doi: 10.1108/JKM-09-2020-0683.

Nederveen Pieterse, A., van Knippenberg, D., & van Ginkel, W. P. Diversity in goal orientation, team reflexivity, and team performance. Organ Behav Hum Dec 2011, 114(2), 153-164. doi:10.1016/j.obhdp.2010.11.003.

Dayan, M., & Basarir, A. Antecedents and consequences of team reflexivity in new product development projects. J Bus Ind Mark 2010, 25(1), 18-29. doi:10.1108/08858621011009128.

Hoegl M, Parboteeah K P. Team reflexivity in innovative projects[J]. R&d Management, 2006, 36(2): 113-125.

Kim T T, Karatepe O M, Lee G. Psychological contract breach and service innovation behavior: psychological capital as a mediator[J]. Service Business, 2018, 12: 305-329.

DeRue D S, Ashford S J. Who will lead and who will follow? A social process of leadership identity construction in organizations[J]. Academy of management review, 2010, 35(4): 627-647.

Narayan, S., Sidhu, J. S., & Volberda, H. W. From attention to action: The influence of cognitive and ideological diversity in top management teams on business model innovation. J Manage Stud 2021, 58(8), 2082-2110. doi: 10.1111/joms.12668.

Sheng, M. L., & Chien, I. Rethinking organizational learning orientation on radical and incremental innovation in high-tech firms. J Bus Res 2016, 69(6), 2302-2308. doi:10.1016/j.jbusres.2015.12.046.

Sinkula, J. M., Baker, W. E., & Noordewier, T. A framework for market-based organizational learning: Linking values, knowledge, and behavior. J Acad Market Sci 1997, 25(4), 305. doi:10.1177/0092070397254003.

Atitumpong, A., & Badir, Y. F. Leader-member exchange, learning orientation and innovative work behavior. J Workplace Learn 2018, 30(1): 32-47. doi: 10.1108/JWL-01-2017-0005.

Chiu, C. Y., Lin, H. C., & Ostroff, C.. Fostering team learning orientation magnitude and strength: Roles of transformational leadership, team personality heterogeneity, and behavioural integration. J Occup Organ Psych 2021, 94(1), 187-216. doi: 10.1111/joop.12333.

Alerasoul, S. A., Afeltra, G., Hakala, H., Minelli, E., & Strozzi, F. Organisational learning, learning organisation, and learning orientation: An integrative review and framework. Hum Resour Manage R 2021, 32(3):100854. doi: 10.1016/j.hrmr.2021.100854.

Patky J. The influence of organizational learning on performance and innovation: a literature review[J]. Journal of Workplace Learning, 2020, 32(3): 229-242.

López S P, Peón J M M, Ordás C J V. Human resource management as a determining factor in organizational learning[J]. Management Learning, 2006, 37(2): 215-239.

Song J H, Chermack T J. A theoretical approach to the organizational knowledge formation process: Integrating the concepts of individual learning and learning organization culture[J]. Human Resource Development Review, 2008, 7(4): 424-442.

Lumpkin G T, Lichtenstein B B. The role of organizational learning in the opportunity–recognition process[J]. Entrepreneurship theory and practice, 2005, 29(4): 451-472.

Fu, N., Flood, P. C., Rousseau, D. M., & Morris, T. Resolving the individual helping and objective job performance dilemma: The moderating effect of team reflexivity. J Bus Res 2021, 129, 236-243. doi: 10.1016/j.jbusres.2021.02.058.

• The relationship between Team prosocial rule breaking and Team reflexivity has not been studied before and auhor has proposed the moderating effect of learning orientation without establiblishing the prime hypothesis between Team prosocial rule breaking and Team reflexivity. I suggested here to first establish and justify the relationship between these two variables with soild theoretical argumentation in separate hypothesis and then check the moderation effect.

Our response: Thanks very much for your helpful suggestions. According to the theoretical framework of this paper, the relationship between team pro-social rule breaking and team reflexivity should not be significant, for the following reasons.

First, team pro-social rule breaking is a behavior that breaks the rules of the organization. Although it is in the interests of the organization, such behavior has certain risks, previous studies have also pointed out that pro-social rule breaking negatively affect the work performance evaluated by others (Morrison, 2006), so it may not directly lead to the emergence of team reflexivity without taking into account the organizational context.

 Second, Following this logic, under the framework of social information processing theory, we believe that the positive relationship between team pro-social rule breaking and team reflexivity can be significant only if a boundary condition effectively eliminate or reduce such risks.

Third, team learning orientation is the team members' consistent perception of the organizational learning atmosphere (Chiu, 2021), a high level of team learning orientation indicates more learning commitment, open-mindedness, and shared vision (Alerasoul, 2021; Patky, 2020). Individuals can perceive the organization's emphasis on learning through the team's daily behavior, and the team's behavioral orientation also implies the organization's tolerance for individual work innovation (López, 2006). Open-mindedness reflects the organization's questioning of routine behaviors and conventions, and the organization is more willing to explore new organizational processes by attempting to overturn old work methods (Lord, 2015). Learning commitment and open-mindedness affect team and individual learning motivation through organizational behavior, and shared vision is the degree to which the organization establishes universally recognized learning-oriented values. A high level of shared vision can promote individuals in the team to form a belief in continuous learning and have a greater impact on the team and individual learning directions. Overall, a high level of team learning orientation means that the team can influence individual learning processes and goals through learning methods. Individuals can perceive a more learning-oriented atmosphere in their interactions with the team, which emphasizes the exploration and acceptance of new things. This atmosphere also has the driving force to improve existing work methods. In this organizational context, pro-social rule breaking behavior may stimulate individuals to think more about the rationality of inherent work processes and attempts, and this attempt is tolerated and encouraged by the organization. Therefore, based on the above discussion, we first construct a moderating relationship, emphasizing the positive impact of team pro-social rule breaking on team reflexivity under the premise of team learning orientation.

Four, similar model assumptions are more common in variables that exhibit both positive and negative aspects, such as perceived overqualification, which reflects an individual's overall assessment of the difference between their own qualifications and organizational requirements. Previous research has confirmed its dual impact on job performance . For instance, Erdogan (2009), in exploring the negative relationship between perceived overqualification and job satisfaction, did not discuss the direct relationship between them. Instead, the author directly proposed the negative effect of the intera

---

## [Decision Letter · Decision Letter 2]

12 Mar 2024

PONE-D-23-17868R2Team pro-social rule breaking and team innovation performance:An information processing theory perspectivePLOS ONE

Dear Dr. Miao,

Thank you for submitting your manuscript to PLOS ONE. After careful consideration, we feel that it has merit but does not fully meet PLOS ONE’s publication criteria as it currently stands. Therefore, we invite you to submit a revised version of the manuscript that addresses the points raised during the review process.

We look forward to receiving your revised manuscript.

Kind regards,

Sharjeel Saleem

Academic Editor

PLOS ONE

Journal Requirements:

Reviewers' comments:

Reviewer's Responses to Questions

**Comments to the Author**

1. If the authors have adequately addressed your comments raised in a previous round of review and you feel that this manuscript is now acceptable for publication, you may indicate that here to bypass the “Comments to the Author” section, enter your conflict of interest statement in the “Confidential to Editor” section, and submit your "Accept" recommendation.

Reviewer #3: (No Response)

2. Is the manuscript technically sound, and do the data support the conclusions?

Reviewer #3: Yes

3. Has the statistical analysis been performed appropriately and rigorously? 

Reviewer #3: Yes

4. Have the authors made all data underlying the findings in their manuscript fully available?

Reviewer #3: No

5. Is the manuscript presented in an intelligible fashion and written in standard English?

Reviewer #3: Yes

6. Review Comments to the Author

Reviewer #3: Thank you for addressing the previous highlighted issues but still I have following concerns.

The Author has ignored my previous observation regarding the direct relationship between

team prosocial rule breaking with innovation performance which hasn’t been checked before. please address this issue by proposing relationship and justify if the results are not significant.

Generally, we have proposed relationship based on supporting theory and previous literature, but your feedback in response letter has demonstrated that the relationship between team pro-social

rule breaking and team reflexivity should not be significant. It means theoretically it is not justifiable so what is the logic behind this framework.

---

## [Author Response · Author response to Decision Letter 2]

27 Mar 2024

Thank you to Reviewer 3 for the valuable guidance on the scientific aspects of the manuscript. In response to your queries, the research team engaged in focused discussions, made modifications to relevant content, and highlighted these changes in the manuscript. We hope that our efforts on the manuscript will assist the reviewer in better understanding the revised content. Below is our response to your inquiries, along with prominent explanations of the manuscript changes.

Reviewer #3: Thank you for addressing the previous highlighted issues but still I have following concerns.The Author has ignored my previous observation regarding the direct relationship between team prosocial rule breaking with innovation performance which hasn’t been checked before. please address this issue by proposing relationship and justify if the results are not significant.

Our response: Thank you for your comments and suggestions on this paper. Based on your suggestions, the research team has proposed the main effect of team pro-social rule breaking on team innovative performance after summarizing the literature on pro-social rule-breaking. The specific content can be found in the following text. Additionally, the modifications related to hypothesis 1 are reflected in the hypothesis formulation, data analysis, and discussion sections of the main text (details are highlighted in yellow). Once again, thank you for your suggestions, which have contributed to enhancing the scientific rigor and logic of this paper.

Hypothetical 1 text:

Researches indicate that pro-social rule breaking, which reflects individuals' autonomy in work and their investment in others' resources and aims to maximize stakeholders' needs, is often related to positive organizational performance [3, 4, 8]. Based on existing research, team pro-social rule breaking may also be a direct factor influencing team innovative performance. 

Firstly, pro-social rule breaking indicates that employees, to a certain extent, overlook organizational norms in pursuit of work efficiency [2]. This behavioral characteristic reflects employees' ability to freely choose more appropriate methods to complete tasks when faced with conflicts between behaviors and norms. Research by Chang (2023) and other scholars has indicated that a high level of work autonomy is often associated with team innovation [41]. At the team level, the overt characteristics of pro-social rule-breaking behavior can facilitate more efficient organization of work interactions among colleagues, which will be beneficial for the improvement of team innovative performance [18]. Secondly, pro-social rule breaking also embodies assistance towards colleagues, manifested as a common practice of helping others within the team [2]. Existing research has shown that helping behavior within teams promotes the enhancement of team innovative performance [19]. Additionally, similar to efficiency-oriented pro-social rule-breaking behavior, pro-social rule-breaking behavior towards customers can convey the true purpose of work to individuals and strengthen individuals' awareness of work autonomy [2]. From an observer's perspective, individuals' perceptions of colleagues' pro-social rule-breaking behavior towards customers can provide specific service inspirations, and these cues can provide external perceptual conditions for individual innovation from aspects such as interaction methods and work processes [9], contributing to the improvement of collective innovative performance at the team level. In conclusion, a hypothesis regarding team pro-social rule-breaking behavior and team innovative performance is proposed: H1: Team pro-social rule-breaking behavior positively influences team innovative performance.

H1: Team pro-social rule-breaking behavior positively influences team innovative performance .

• Generally, we have proposed relationship based on supporting theory and previous literature, but your feedback in response letter has demonstrated that the relationship between team pro-social rule breaking and team reflexivity should not be significant. It means theoretically it is not justifiable so what is the logic behind this framework.

Our response: Thanks for your feedback and suggestions on this paper. Based on the theoretical analysis framework of this paper, there is a causal path relationship between team pro-social rule-breaking behavior and team reflexivity, but it requires situational conditions to be triggered. Reviewing similar existing studies, we believe that team pro-social rule breaking needs to be stimulated by the variable of team learning orientation to positively influence team reflexivity and thus enhance team innovative performance. In other words, team pro-social rule breaking needs specific organizational factors to trigger a tendency towards cognitive change among team members.

In the field of organizational science, research concepts with similar double-edged sword effects generally exhibit such characteristics, such as pro-social rule breaking behavior (which combines altruism and rule-breaking characteristics) and perceived overqualification (POQ, where employees perceive their qualifications to exceed the experiential, educational, and skill requirements of their position, potentially leading to higher job performance but also lower job performance due to misalignment, perceived unfairness, and perceived deprivation mechanisms that may even lead to organizational indignation). Studies on these similar variables indicate that the reasons for the insignificant path relationship between the independent variable of pro-social rule-breaking behavior and the mediating and outcome variables can be divided into two types. The first type is when the independent variable is indeed unrelated to the mediating variable, meaning there is no correlation between the two; the second type is when the independent variable's impact on the mediating variable is multifaceted, meaning there may be both positive and negative effects coexisting. At the data level, these effects may cancel each other out, resulting in no correlation, and only under certain interactive situational factors can the positive effects "emerge."

Shum (2022) conducted a study on the impact of employees' pro-social safety rule breaking on customer satisfaction. Based on practical theory, the study indicates that employees' pro-social safety violations only have a negative impact on bystanders' service performance evaluations and perceptions of safety when considering the customer's role. In the context of the COVID-19 pandemic, customers' awareness of their role has strengthened their expectations for self-service safety and service quality. Pro-social safety violations, from the bystander's perspective, manifest as service personnel breaking hotel rules, potentially endangering the safety interests of other guests represented by bystanders. Therefore, this behavior can lead to negative evaluations of hotel service performance by bystanders and raise doubts about service safety. 

The same method of hypothesis presentation is also found in other research areas within organizational behavior studies. Taking POQ as an example, Lobene (2013) highlighted that the interaction effect between mission orientation and POQ negatively impacts turnover intention. When employees have a higher sense of work mission, the negative impact of POQ on turnover intention diminishes. Cheng (2018), based on equity theory, validated the interactive effect between achievement needs and POQ on harmonious passion. Employees with a high need for achievement may be more susceptible to the mismatch between individuals and their work (i.e., being overqualified for their position), leading to a loss of harmonious passion for work. 

Deng (2018) directly proposed in a study on the impact of POQ on organizational performance that the interaction effect between interpersonal influence and POQ affects social acceptance. The research findings indicate that the level of interpersonal influence determines the relationship between POQ and social acceptance. When interpersonal influence is high, this relationship is positive; when interpersonal influence is low, this relationship is negative. The data analysis results of two time-lagged studies support this hypothesis, and the examination of correlation confirms that the correlation between POQ and social acceptance is not significant.

Erdogan (2020) pointed out in his study on the impact of POQ on Advice Network Centrality that the moderating effect of P-O fit on the relationship between POQ and Advice Network Centrality only has a significant impact when the P-O fit moderating factor is present. Specifically, when P-O fit is high, the moderating effect has a positive impact on the mediating factor, whereas when P-O fit is low, the moderating effect has a negative impact on the mediating factor. Data analysis results indicate that the correlation between POQ and giving advice is not significant, but the moderating effect of P-O fit on the relationship between POQ and giving advice is significant. 

As mentioned earlier, similar to POQ, the impact of pro-social rule-breaking behavior on organizations and individuals is diverse. In the data analysis phase, it may be observed that the correlation between the independent variable (PSRB) and the mediating variable is not significant. The influence on the mediating variable is achieved only under specific contextual factors. Therefore, drawing from relevant literature, the research team directly proposes the interactive effect of TPSRB and team learning orientation on positive team reflexivity. Additionally, considering the completeness of the paper, based on your suggestion, the research team has added a discussion on the correlation between TPSRB and team reflexivity in the discussion section to enhance the logical coherence of the paper.

Once again, thank you for your suggestions, which have improved the logical flow of this paper.

References:

CAI Y, CHENG J, LI J. Rules can maintain harmony? The influence of team pro-social rule breaking climate on team performance from the perspective of harmony management[J]. Acta Psychologica Sinica, 2022, 54(1): 66.

Shum C, Ghosh A. Safety or service? Effects of employee prosocial safety-rule-breaking on consumer satisfaction[J]. International Journal of Hospitality Management, 2022, 103: 103225.

Lobene E V, Meade A W. The effects of career calling and perceived overqualification on work outcomes for primary and secondary school teachers[J]. Journal of Career Development, 2013, 40(6): 508-530.

Deng H, Guan Y, Wu C H, et al. A relational model of perceived overqualification: The moderating role of interpersonal influence on social acceptance[J]. Journal of Management, 2018, 44(8): 3288-3310.

Erdogan B, Karaeminogullari A, Bauer T N, et al. Perceived overqualification at work: Implications for extra-role behaviors and advice network centrality[J]. Journal of Management, 2020, 46(4): 583-606.

---

## [Decision Letter · Decision Letter 3]

1 May 2024

Team pro-social rule breaking and team innovation performance:An information processing theory perspective

PONE-D-23-17868R3

Dear Dr. Miao,

We’re pleased to inform you that your manuscript has been judged scientifically suitable for publication and will be formally accepted for publication once it meets all outstanding technical requirements.

Kind regards,

Sharjeel Saleem

Academic Editor

PLOS ONE

Additional Editor Comments (optional):

Reviewers' comments:

Reviewer's Responses to Questions

**Comments to the Author**

1. If the authors have adequately addressed your comments raised in a previous round of review and you feel that this manuscript is now acceptable for publication, you may indicate that here to bypass the “Comments to the Author” section, enter your conflict of interest statement in the “Confidential to Editor” section, and submit your "Accept" recommendation.

Reviewer #3: (No Response)

2. Is the manuscript technically sound, and do the data support the conclusions?

Reviewer #3: Partly

3. Has the statistical analysis been performed appropriately and rigorously? 

Reviewer #3: Yes

4. Have the authors made all data underlying the findings in their manuscript fully available?

Reviewer #3: Yes

5. Is the manuscript presented in an intelligible fashion and written in standard English?

Reviewer #3: Yes

6. Review Comments to the Author

Reviewer #3: The Revised manuscript has incorporated suggestions. .
